# TRUST OR ESCALATE: LLM JUDGES WITH PROVABLE GUARANTEES FOR HUMAN AGREEMENT

**Jaehun Jung**[1]    **Faeze Brahman**[1 2]    **Yejin Choi**[1 2]

[1]University of Washington      [2]Allen Institute for Artificial Intelligence

## ABSTRACT

We present a principled approach to provide LLM-based evaluation with a rigorous guarantee of human agreement. We first propose that a reliable evaluation method should not uncritically rely on model preferences for pairwise evaluation, but rather assess the confidence of judge models and selectively decide when to trust its judgement. We then show that under this *selective evaluation* framework, human agreement can be provably guaranteed—such that the model evaluation aligns with that of humans to a user-specified agreement level. As part of our framework, we also introduce *Simulated Annotators*, a novel confidence estimation method that significantly improves judge calibration and thus enables high coverage of evaluated instances. Finally, we propose *Cascaded Selective Evaluation*, where we use cheaper models as initial judges and escalate to stronger models only when necessary—again, while still providing a provable guarantee of human agreement. Experimental results show that *Cascaded Selective Evaluation* guarantees strong alignment with humans, far beyond what LLM judges could achieve without selective evaluation. For example, on a subset of Chatbot Arena where GPT-4 almost never achieves 80% human agreement, our method, even while employing substantially cost-effective models such as Mistral-7B, *guarantees* over 80% human agreement with almost 80% test coverage. 🎧

## 1 INTRODUCTION

Imagine we need to evaluate 1 million pairs of model generations—a task whose scale makes human annotation impractical, if not impossible. Today, a commonly proposed solution is to 'just ask GPT-4'(Zheng et al., 2023; Dubois et al., 2023), realizing a tempting idea that large language models (LLMs) may serve as a scalable substitute for manual annotation (Chiang & Lee, 2023). However, this compelling prospect comes with a crucial caveat—LLM-based evaluation would always remain, at best, an approximation of human judgement. Without a provable guarantee of reliability, it is no surprise that the judge model has to be chosen heuristically, often times to be the strongest and the most expensive model available (*e.g.,* GPT-4). Yet, prior works show that even the strongest judge models suffer from systematic biases (Wang et al., 2023; Thakur et al., 2024) and over-confidence (Xiong et al., 2024), casting doubt on the dependability of these models. This raises a fundamental question: *How can we guarantee the reliability of LLM-based evaluation?*

In this work, we aim to improve the reliability of LLM-based evaluation by providing a rigorous guarantee of human agreement. That is, given a user-defined risk level $\alpha$, we provide a guarantee that, for an unseen instance $x$,

$$P(\textit{LLM preference on } x \textit{ agrees with human} \mid \textit{LLM evaluates } x) \geq 1 - \alpha.$$

To provide this guarantee, we posit that a reliable evaluation framework should not only consider the preference of a model, but also the validity of the preference—*i.e.,* how likely humans would agree with the model judgement. When a model cannot confidently evaluate a given instance, we should not rely on its evaluated result. This motivates *selective evaluation*: we evaluate an instance with an LLM judge, assess the confidence that humans would agree with its evaluation, then decide whether or not to trust the evaluated result. We show that under this framework, human agreement can indeed be guaranteed—both theoretically and empirically—by choosing when to trust the model via *fixed sequence testing* (Bauer, 1991) on a small calibration set.

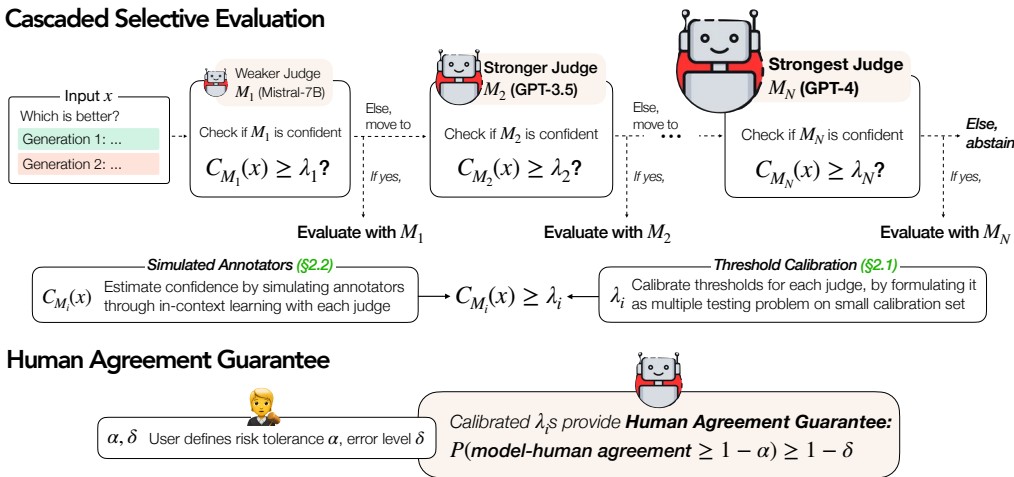

Figure 1: Illustration of Cascaded Selective Evaluation. We start with a small, cost-effective model as initial judge, estimate its confidence, and escalate to a stronger model only when the previous judge is not confident. By calibrating when to trust which judge model, **our method provides a rigorous guarantee of human agreement while employing substantially cheaper judge models.**

The practicality of selective evaluation lies not only in achieving high agreement with humans, but also in maximizing the coverage of evaluated instances without abstention—a factor that depends on the quality of confidence measure. We find that existing methods for confidence estimation (*e.g.,* predictive probability) are brittle even with the strongest judge model, as they tend to overestimate human agreement. We then propose **Simulated Annotators**, a novel method to simulate diverse annotator preferences through in-context learning and estimate confidence as an agreement ratio between the simulations. Without relying on any external supervision, *Simulated Annotators* significantly improves both the calibration and failure prediction of LLM judges. As a result, selective evaluation can be done with high coverage while satisfying the prescribed human agreement level.

Moreover, since our framework provides a model-agnostic guarantee of human agreement, we no longer have to rely solely on GPT-4 for evaluation. We propose **Cascaded Selective Evaluation** (Figure 1), where we start from a substantially cheaper LM (*e.g.,* Mistral-7B) as a judge, and escalate to a stronger model only when the previous judge is not sufficiently confident—all while guaranteeing high agreement with humans. Importantly, users do not have to manually choose when to use which judge model; given a user-specified risk tolerance, the abstention policy is automatically decided to maintain risk control.

We test our method across preference domains including summarization and real-world user-chatbot interaction, and find that *Cascaded Selective Evaluation* significantly reduces the evaluation overhead while guaranteeing high agreement. For example, our method can outperform GPT-4 by achieving over 80% human agreement in ChatArena (Li et al., 2024a), while covering 79.1% of all samples, among which 88.1% are evaluated by substantially cheaper Mistral-7B or GPT-3.5 instead of GPT-4. We also show that our abstention policy closely aligns with the subjectivity perceived by humans, rather than relying on shallow features such as length ratio or token overlap. Overall, our work suggests a principled approach to make LLM-based evaluation more reliable yet cost-effective, without exclusively counting on the capabilities of the most advanced LLMs as judges.

## 2  CASCADED SELECTIVE EVALUATION

When performing pairwise evaluation with LLMs, we want a type of guarantee that the model agrees with the majority of human annotators. To realize this guarantee, we propose *selective evaluation*, a framework that employs an abstention policy to decide whether an LLM is sufficiently confident to evaluate an instance. More formally, let $f_{LM} : \mathcal{X} \rightarrow \mathcal{Y}$ denote the LLM judge, where the input $x \in \mathcal{X}$ consists of a query $q$ and a pair of generations $(a_1, a_2)$, and the output $y \in \mathcal{Y}$ is a preference label between $a_1$ and $a_2$ (*e.g.,* $a_1 \succ a_2$). Introducing a confidence measure $c_{LM} : \mathcal{X} \rightarrow [0, 1]$, we define selective evaluator as:

$$(f_{LM}, c_{LM})(x) = \begin{cases} f_{LM}(x) & \text{if } c_{LM}(x) \geq \lambda, \\ \emptyset & \text{otherwise.} \end{cases} \quad (1)$$

An example of $c_{LM}$ is the probability assigned by $f_{LM}$ to its predicted label (predictive probability), a popular choice of confidence measure in selective classification (Geifman & El-Yaniv, 2017). $\lambda$ is a hyperparameter that trades off the precision (*i.e.,* the accuracy of evaluator aligning with human judgements) against the coverage (*i.e.,* the ratio of instances evaluated without abstention). The key advantage of selective evaluation is that by calibrating $\lambda$ in a principled manner, we can provide a rigorous guarantee of human agreement while maintaining high coverage. That is, given a user-defined risk tolerance $\alpha$ and an error level $\delta$, one can provably guarantee that

$$P(f_{LM}(x) = y_{human}|c_{LM}(x) \geq \lambda) \geq 1 - \alpha \tag{2}$$

is satisfied with probability at least $1 - \delta$. In the following sections, we illustrate how to search for $\widehat{\lambda}$ that satisfies this guarantee (§2.1), how to define a good confidence measure $c_{LM}$ (§2.2), and how to extend selective evaluation from a single model to cascades of judge models (§2.3).

## 2.1 PROVIDING HUMAN AGREEMENT GUARANTEE

Our human agreement guarantee can be satisfied by formulating selection of $\lambda$ as a multiple hypothesis testing problem (Bates et al., 2021; Angelopoulos et al., 2022). Specifically, given access to a small calibration set $D_{cal} \sim P(x, y_{human})$ of human preferences[1], we can measure an empirical risk $\widehat{R}(\lambda)$ of disagreeing with humans when using a threshold $\lambda$:

$$\widehat{R}(\lambda) = \frac{1}{n(\lambda)} \sum_{(x, y_{human}) \in D_{cal}} \mathbb{1}\{f_{LM}(x) \neq y_{human} \wedge c_{LM}(x) \geq \lambda\}, \tag{3}$$

where $n(\lambda) := \sum_{(x, y_{human}) \in D_{cal}} \mathbb{1}\{c_{LM}(x) \geq \lambda\}$. Since the empirical risk is a binomial random variable with $n(\lambda)$ trials, we can compute the exact $(1 - \delta)$ upper confidence bound of the risk as:

$$\widehat{R}^+(\lambda) = \sup \left\{ R : P(\text{Bin}(n(\lambda), R) \leq \lceil n(\lambda)\widehat{R}(\lambda)\rceil) \geq \delta \right\}. \tag{4}$$

Note here that the risk is near-monotonic, *i.e.,* it tends to increase as $\lambda$ decreases. This allows us to use fixed sequence testing (Bauer, 1991), wherein we test from the largest value of $\lambda$ (*e.g.,* 0.999) to a progressively smaller value, and stop at the last time $\widehat{R}^+(\lambda)$ is below the target risk $\alpha$.

$$\widehat{\lambda} = \inf \left\{ \lambda : \widehat{R}^+(\lambda') \leq \alpha \text{ for } \forall \lambda' \geq \lambda \right\}. \tag{5}$$

**Theorem 1** *Consider a threshold $\widehat{\lambda}$ chosen as above, and a selective evaluator $(f_{LM}, c_{LM})$ operating based on $\widehat{\lambda}$. Then, Equation (2) is satisfied with probability at least $1 - \delta$.*

We present the proof in §A.1. Our test procedure resembles that from selective classification literature (Geifman & El-Yaniv, 2017), but we adopt fixed-sequence testing instead of Bonferroni correction, which may be too conservative for a large hypothesis space. Compared to recent works on risk control for LLMs, we provide exact, tighter bound on the selective risk (instead of approximating it; Yadkori et al. 2024), and guarantee with high probability that the risk is below $\alpha$ conditional on the calibration data (as opposed to being marginally centered at $\alpha$; Gui et al. 2024).

## 2.2 SIMULATED ANNOTATORS

While human agreement guarantee can be met with any choice of (near-monotonic) confidence measure, the coverage of selective evaluation essentially depends on how good this measure is— *i.e.,* whether $c_{LM}$ truly reflects if humans would agree with LLM evaluation. In this section, we first test out popular confidence estimation methods for LLMs and show that they fail to accurately represent model uncertainty. We then introduce *Simulated Annotators* as a promising alternative.

**Existing Methods.** We first consider two types of existing confidence measure[2]—(1) *predictive probability*: as the most straightforward proxy of confidence, we use the likelihood of preference

---

[1]When we have multiple human annotation $y_i$s per input $x$, we define $y_{human} := \arg\max_y \sum_i \mathbb{1}\{y_i = y\}$.

[2]We also consider more sophisticated methods (*e.g.,* sampling chain-of-thoughts and estimating their *semantic entropy*; Kuhn et al. 2023) in §B, but find their performance to be mostly on-par with the above methods, despite the increased cost.

Table 1: Performance of confidence measures across judge models. **Simulated Annotators consistently outperforms baselines both in calibration and failure prediction, especially improving the reliability of weaker judge models (*GPT-3.5-turbo* and *Mistral-7B*).**

| Dataset | | AlpacaEval | | | | TL;DR | | | |
| --- | --- | --- | --- | --- | --- | --- | --- | --- | --- |
| Method | | Acc. | ECE↓ | AUROC | AUPRC | Acc. | ECE↓ | AUROC | AUPRC |
| *GPT-4-turbo* | Predictive Probability | 0.724 | 0.217 | 0.642 | 0.852 | 0.760 | 0.196 | 0.731 | 0.890 |
| | Verbalized Confidence | 0.724 | 0.215 | 0.550 | 0.774 | 0.760 | 0.194 | 0.548 | 0.792 |
| | Randomized Annotators | 0.720 | 0.113 | 0.705 | 0.866 | 0.779 | 0.079 | 0.734 | 0.905 |
| | Simulated Annotators (Maj.) | 0.730 | 0.106 | 0.718 | 0.873 | 0.783 | 0.062 | **0.755** | **0.921** |
| | Simulated Annotators (Ind.) | **0.734** | **0.095** | **0.723** | **0.877** | **0.788** | **0.039** | 0.755 | 0.921 |
| *GPT-3.5-turbo* | Predictive Probability | 0.644 | 0.293 | 0.581 | 0.691 | 0.667 | 0.228 | 0.653 | 0.786 |
| | Verbalized Confidence | 0.644 | 0.306 | 0.505 | 0.595 | 0.667 | 0.211 | 0.607 | 0.716 |
| | Simulated Annotators (Ind.) | **0.694** | **0.058** | **0.632** | **0.793** | **0.725** | **0.043** | **0.704** | **0.842** |
| *Mistral-7B-it* | Predictive Probability | 0.618 | 0.374 | 0.457 | 0.579 | 0.661 | 0.306 | 0.613 | 0.735 |
| | Verbalized Confidence | 0.618 | 0.414 | 0.490 | 0.627 | 0.661 | 0.335 | 0.578 | 0.680 |
| | Simulated Annotators (Ind.) | **0.684** | **0.075** | **0.632** | **0.772** | **0.696** | **0.103** | **0.654** | **0.807** |

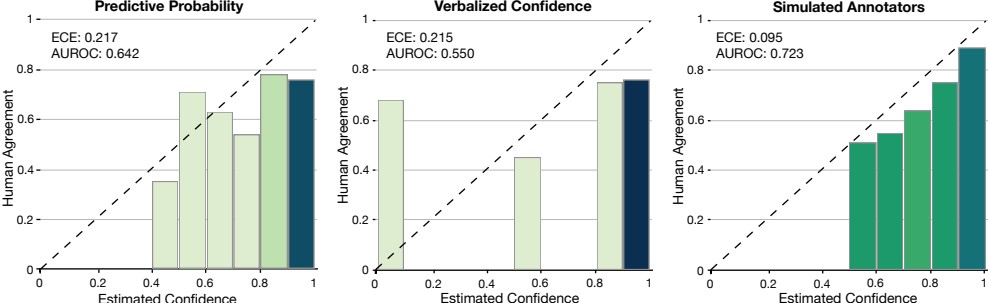

Figure 2: Reliability plot for confidence estimation methods, using GPT-4 as judge on AlpacaEval. Dashed lines denote perfect calibration, and darker bars denote more samples in the corresponding bins. **Simulated Annotators reduces expected calibration error by 50% compared to the baselines, mitigating over-confidence** observed in predictive probability and verbalized confidence.

label predicted by the LLM judge, *i.e.,* $c_{LM}(x) = \max_y p_{LM}(y|x)$; (2) *verbalized confidence*: following Tian et al. (2023b), we directly prompt the LLM judge to express its confidence in a scalar value. We evaluate these methods in terms of the expected calibration error (Naeini et al., 2015), AUROC and AUPRC, using the non-tied instances in two standard benchmarks: AlpacaEval (Dubois et al., 2023) for open-domain chat assistant and TL;DR (Stiennon et al., 2020) for summarization.

**Canonical methods overestimate human agreement.** The results are shown in Table 1 and Figure 2 (left, middle). Unlike prior reports that the canonical methods work well for tasks with small label space (Kadavath et al., 2022; Tian et al., 2023b), they consistently lead to over-confidence when used for preference evaluation. Notably, the results are pronounced even with the strongest LLM judge *GPT-4-turbo*, although its agreement with human majority is known to be comparable to an average human annotator (Sottana et al., 2023; Zheng et al., 2023).

The above results suggest that simply achieving human-level performance may not be sufficient for reliable evaluation: while an LLM judge can be as accurate as a single human annotator, it tends to be over-confident in estimating its agreement with the majority of annotators. This contrasts with the standard practice in human evaluation, which involves collecting multiple annotations per instance and assessing the level of agreement between them; the evaluation is deemed reliable only when there is high inter-annotator agreement. Motivated by this discrepancy, we derive *Simulated Annotators*, a confidence measure that simulates diverse annotator preferences with in-context learning. Concretely, given $K$ (*e.g.,* 3) examples of preference annotations per $N$ (*e.g.,* 5) human annotators, we simulate annotators by $K$-shot prompting the model for $N$ times and ensemble the results:

$$c_{LM}(x) = \max_y \frac{1}{N} \sum_{j=1}^{N} p_{LM}(y|x; (x_{1,j}, y_{1,j}), \cdots, (x_{K,j}, y_{K,j})),$$

where $(x_{i,j}, y_{i,j})$ is the $i$-th in-context example from the $j$-th annotator. Likewise, the judge prediction $f_{LM}(x)$ is defined as $\arg\max_y \sum_{j=1}^{N} p_{LM}(y|x; (x_{1,j}, y_{1,j}), \cdots, (x_{K,j}, y_{K,j}))$. Intuitively,

---

**Algorithm 1** Cascaded Selective Evaluation

---

**Input:** A list of judges $\mathcal{M} = (M_1, \cdots, M_{|\mathcal{M}|})$, a calibration set $D_{cal}$ and test set $D_{test}$ to be evaluated, risk tolerance $\alpha$ and error level $\delta$
**Output:** A set of evaluated results $S$
    $\Lambda \leftarrow \text{calibrate}(\mathcal{M}, D_{cal}, \alpha, \delta)$         ▷ Calibrate thresholds $\lambda \in \Lambda$ on $D_{cal}$ (§A.2).
    $S \leftarrow \emptyset$         ▷ Initialize a set of evaluation results.
    **for** $x \in D_{test}$ **do**
        **for** $i = 1$ to $|\mathcal{M}|$ **do**         ▷ Iterate through the cascade of judge models.
            **if** $c_{M_i}(x) \geq \lambda_i$ **then**         ▷ Evaluate $x$ only when $M_i$ is sufficiently confident.
                $S \leftarrow S \cup \{(x, f_{M_i}(x))\}$
            **break**
    **return** $S$         ▷ Return the evaluated results.

---

the confidence $c_{LM}$ becomes low when multiple simulated annotators disagree with each other. We typically set $K, N \leq 5$, and ablate the effect of number of simulated annotators in §3.5.

**Simulated Annotators improves reliability, even for weaker judge models.** The results with $K = N = 5$ are shown in Table 1 (*Simulated Annotators (Ind.)*) and Figure 2 (right). Simulated Annotators significantly outperforms popular confidence measures, reducing ECE by 50% and improving AUROC by 13% for GPT-4. Surprisingly, our method improves the reliability of even the weaker judge models—while they do underperform GPT-4 in accuracy, their estimated confidence is on-par or even better than GPT-4 when using the baseline confidence measures.

Despite the substantial performance gain in Simulated Annotators, it remains unclear whether the gain truly comes from simulating diverse human preferences. We analyze this using two ablations on the few-shot examples given to the LLM judge: (1) *randomized annotators*: using the same set of inputs $x_{i,j}$ but random-assigning labels $y_{i,j} \sim \text{Ber}(0.5)$, and (2) *simulated annotators (majority)*: using $(x_{i,j}, y_{i,human})$ where $y_{i,human}$ is the majority preference of human annotators given input $x_{i,j}$.[3] We fix $K = 5$ and $N = 5$ for all ablations. As shown in Table 1, *Simulated Annotators (Maj.)* is consistently better than *Randomized Annotators*, but slightly underperforms *Simulated Annotators (Ind.)* that models individual preference. The performance of majority-based Simulated Annotators, however, is encouraging, as the method can be applied to cases where we do not have access to multiple human annotations per each instance $x$. Overall, the result demonstrates that simulating diverse annotator preferences is helpful, and even in the absence of such data, our method improves the reliability of LLM judges over the existing methods.

## 2.3 CASCADING SELECTIVE EVALUATORS

The strong performance of Simulated Annotators demonstrates that even the weaker LLM judges—despite not as accurate as a larger judge—may accurately predict when they are likely to agree with human annotators. Leveraging this finding, we propose **Cascaded Selective Evaluation**, as illustrated in Figure 1 and formalized in Algorithm 1. Given a list of judge models $\mathcal{M}$, we start with a weaker yet cheaper model as an evaluator, and only when the model is not sufficiently confident, we iteratively move on to a stronger model. Notably, the confidence threshold $\lambda$ for each judge model can be chosen following the same process as in §2.1, providing the guarantee of risk control across the cascades of models (see §A.2 for details). This way, selective evaluation can operate at a significantly lower cost than using the strongest model from the start, while still maintaining a rigorous guarantee of human agreement.

## 3 EXPERIMENTAL RESULTS

### 3.1 EVALUATING GENERATED SUMMARIES

**Experimental Setup.** We first test our approach for evaluating summaries on TL;DR dataset (Stiennon et al., 2020). We use a cascade of *Mistral-7B-instruct-v0.2* (Jiang et al., 2023), *GPT-3.5-turbo*

---

[3]Here, as each input instance is associated with a single majority label $y_{human}$, we induce difference between simulations by using different set of inputs $x_{i,j}$ per simulated annotator.

Table 2: Comparison against baselines on TL;DR, with target agreement level $1 - \alpha = 0.9$. The results are averaged across 1000 runs with random data split. Guarantee Success Rate is defined as the ratio of successful runs that achieve empirical human agreement larger than or equal to $1 - \alpha$. **Cascaded Selective Evaluation is the only method that achieves high guarantee success rate, while maintaining high coverage.**

| Method | Evaluator Composition (%) | | | Coverage (%) | Guarantee Success Rate (%) |
|---|---|---|---|---|---|
| | *Mistral-7B* | *GPT-3.5-turbo* | *GPT-4-turbo* | | |
| No Selection | 0.0 | 0.0 | 100.0 | 100.0 | 0.0 |
| Heuristic Selection | 0.0 | 0.0 | 100.0 | 89.6 | 42.0 |
| Cascaded Heuristic Selection | 59.6 | 15.0 | 25.5 | 64.6 | 0.0 |
| | 100.0 | 0.0 | 0.0 | 5.6 | 54.7 |
| Point-Estimate Calibration | 0.0 | 100.0 | 0.0 | 9.4 | 79.0 |
| | 0.0 | 0.0 | 100.0 | 57.7 | 47.5 |
| **Cascaded Selective Evaluation** | **28.3** | **28.2** | **43.5** | **55.7** | **90.8** |

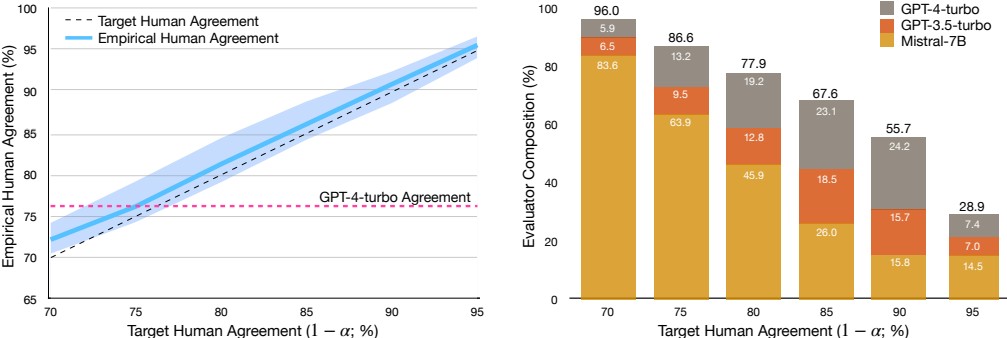

Figure 3: TL;DR results. **Cascaded Selective Evaluation guarantees human agreement far beyond a level achievable by GPT-4 without abstention (Left), while employing substantially weaker judge models (Right).** Solid blue line denotes average human agreement over 1000 runs on the dataset, and the light blue region denotes the min / max agreement within the 1000 runs.

and *GPT-4-turbo* (Achiam et al., 2023) as judges. Observing that the dataset provides multiple human annotations per input, we use *Simulated Annotators (Ind.)* with $K = N = 5$. We fix the size of calibration set $|D_{cal}| = 500$ and $\delta = 0.1$, and run the experiments for 1000 random splits of calibration and test set. For baselines, we consider: (1) Heuristic Selection: using GPT-4 as a judge and setting $\lambda = 1 - \alpha$, assuming perfect calibration; (2) Cascaded Heuristic Selection: a variant of Heuristic Selection using the same cascades of judge models as ours; (3) Point-Estimate Calibration: setting $\lambda$ as the smallest value that satisfies $\widehat{R}(\lambda) \leq \alpha$ in $D_{cal}$, without hypothesis testing.

In Figure 3, we show that human agreement guarantee is satisfied with our approach across all levels of target human agreement, far beyond what GPT-4 can achieve without abstention. Notably, unlike prior works (Gui et al., 2024; Mohri & Hashimoto, 2023) that only controls the risk in expectation over calibration sets (solid blue line), our method guarantees with high probability that each individual run would satisfy the target agreement level (light blue region). Moreover, as shown in right plot, the high agreements can be achieved while the majority of evaluation are done with substantially smaller LLMs than GPT-4. For example, our method can outperform GPT-4 with 80% human agreement, while 75% of the evaluations are done by Mistral-7B or GPT-3.5.

We also compare against selective baselines in Table 2, in terms of their coverage and guarantee success rate. All baselines fail to provide meaningful guarantee success rate without significantly sacrificing the coverage. This includes Point-Estimate Calibration, which makes use of the test statistics in calibration data (Ramirez et al., 2024). On the contrary, Cascaded Selective Evaluation achieves over 90% success rate—just as expected by setting $\delta = 0.1$—attesting to its reliability.

## 3.2 EVALUATING LLM-BASED CHAT ASSISTANTS

**Experimental Setup.** Next, we test our approach for evaluating general-purpose LLM assistants on two datasets: Chat(bot) Arena (Zheng et al., 2023) with real-world user-assistant interaction[4]

---

[4]We use an evaluation set with 5.2k instances collected by Li et al. (2024a).

Table 3: Comparison to baselines on ChatArena, with target agreement level $1 - \alpha = 0.85$. The results are averaged across 1000 runs with random data split. Consistent with results on TL;DR, **our method successfully guarantees target agreement level while maintaining high coverage.**

| Method | Evaluator Composition (%) | | | Coverage (%) | Guarantee Success Rate (%) |
|---|---|---|---|---|---|
| | *Mistral-7B* | *GPT-3.5-turbo* | *GPT-4-turbo* | | |
| No Selection | 0.0 | 0.0 | 100.0 | 100.0 | 0.0 |
| Heuristic Selection | 0.0 | 0.0 | 100.0 | 95.2 | 0.1 |
| Cascaded Heuristic Selection | 57.1 | 15.2 | 27.7 | 79.7 | 0.3 |
| | 100.0 | 0.0 | 0.0 | 0.0 | 0.0 |
| Point-Estimate Calibration | 0.0 | 100.0 | 0.0 | 40.5 | 57.2 |
| | 0.0 | 0.0 | 100.0 | 60.9 | 54.4 |
| **Cascaded Selective Evaluation** | **23.7** | **58.8** | **17.5** | **63.2** | **91.0** |

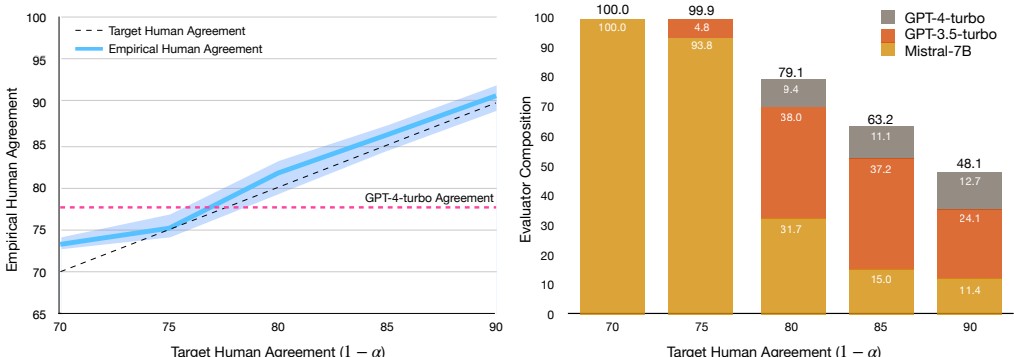

Figure 4: ChatArena results. **Our approach guarantees target human agreement level (Left) while majority of evaluations are done with weaker judge models, Mistral-7B and GPT-3.5 (Right).**

and Auto-J (Li et al., 2023), a curated benchmark for meta-evaluation of LLM judges. We employ the same cascades of models as in §3.1, but this time we use *Simulated Annotators (Maj.)*, as both datasets only provide one human annotation per input $x$. We set $K = N = 5$ and $|D_{cal}| = 500$ for ChatArena, and $K = N = 3$, $|D_{cal}| = 392$ for Auto-J, considering the small size of the benchmark.

The results are shown in Figure 4 and Table 3 for ChatArena, and in Figure 5 and Table 8 for Auto-J. Again, we confirm that human agreement guarantee can be achieved across all levels of $\alpha$. On ChatArena, unlike Point-Estimate Calibration with GPT-4 whose guarantee success rate is below 60%, our method achieves 91% guarantee success rate while only using GPT-4 for 17.5% of the evaluations. The performance is particularly pronounced in Auto-J where GPT-4 without abstention could only achieve 63.2% agreement with humans (Figure 5), potentially due to the fact that the dataset introduces additional *tie* label unlike the other two datasets. In stark contrast, Cascaded Selective Evaluation guarantees up to 80% human agreement with high probability.

Next, we conduct further ablations and analyses to better understand the working of our method.

## 3.3 UNDERSTANDING ABSTENTION POLICY

When does the model abstain from evaluation? One concern with selective evaluation is that its abstention policy may not align with the human-perceived subjectivity of each instance and instead rely on shallow heuristics, *e.g.,* choosing to abstain when the pair of generations have large token overlap. To address this concern, we analyze whether there exists a significant difference in human-perceived subjectivity between model-abstained samples and evaluated samples. We first collect 3-5 additional human annotations per each instance in ChatArena, and measure the

Table 4: Comparison between abstained vs. evaluated samples. **Our abstention policy aligns with how humans agree with each other (IAA)**, exhibiting no significant reliance on shallow heuristics (length ratio, token overlap).

| Dimension | Abstained Samples | Evaluated Samples |
|---|---|---|
| Human IAA | 0.815 (0.031) | 0.902 (0.025) |
| Length Ratio | 0.242 (0.014) | 0.245 (0.025) |
| Token Overlap | 0.623 (0.049) | 0.592 (0.054) |

Table 5: Evaluation under distribution shift on ChatArena. We induce distribution shift by sampling the calibration and test set respectively from two disjoint sets of instances with no overlap of evaluated models. We iterate experiments for 1000 random splits and aggregate the results. **Cascaded Selective Evaluation guarantees high agreement even under the realistic distribution shift.**

| Target Human Agreement (%) | Empirical Human Agreement (%) | Coverage (%) | Guarantee Success Rate (%) |
|---|---|---|---|
| 70.0 | 73.4 | 100.0 | 100.0 |
| 75.0 | 75.3 | 91.4 | 92.5 |
| 80.0 | 80.8 | 72.1 | 90.8 |
| 85.0 | 85.2 | 55.4 | 91.0 |
| 90.0 | 90.1 | 31.8 | 90.7 |

Table 6: Impact of number of simulated annotators $N$ on ChatArena, with $1 - \alpha = 0.85$. Larger number of simulations generally leads to better coverage, while human agreement is guaranteed even with a small $N$. **For all values of $N$, Cascaded Selective Evaluation guarantees high agreement with humans while reducing the API cost by 40% compared to GPT-4 without abstention.**

| Method | Empirical Human Agreement (%) | Coverage (%) | Guarantee Success Rate (%) | Relative API Cost |
|---|---|---|---|---|
| GPT-4 ($N = 1$) | 77.8 | 100.0 | 0.0 | 1.000 |
| Cascaded Selective Evaluation ($N = 1$) | 85.2 | 60.9 | 90.3 | 0.655 |
| GPT-4 ($N = 2$) | 78.2 | 100.0 | 0.0 | 2.000 |
| Cascaded Selective Evaluation ($N = 2$) | 85.7 | 61.5 | 90.8 | 1.288 |
| GPT-4 ($N = 3$) | 78.1 | 100.0 | 0.0 | 3.000 |
| Cascaded Selective Evaluation ($N = 3$) | 85.5 | 62.1 | 90.3 | 1.920 |
| GPT-4 ($N = 5$) | 78.5 | 100.0 | 0.0 | 5.000 |
| Cascaded Selective Evaluation ($N = 5$) | 85.8 | 63.2 | 91.0 | 2.849 |

inter-annotator agreement (IAA) as a proxy of human-perceived subjectivity[5]. Then, we compare the difference in IAA between abstained and evaluated samples when the target agreement level is set to 0.9. We also measure the difference in terms of shallow features, specifically the pairwise length ratio and the token-overlap (ROUGE-L) within each instance. For further details, see §C.2.

Table 4 presents the results. The average inter-annotator agreement is 0.815 ($\sigma^2 = 0.031$) for abstained samples and 0.902 ($\sigma^2 = 0.025$) for evaluated samples, a statistically significant difference in two-sample t-test with $p < 1e-8$. This is in contrast with both the length ratio and token overlap, for which the differences between the two sets are not significant ($p > 0.10$). In fact, for token overlap, the abstained examples exhibit higher ROUGE-L on average than the evaluated samples. Overall, these results show that the instances abstained by LLM judges tend to be more subjective even for humans (with no evidence of reliance on some spurious heuristics), suggesting that the confidence elicited by Simulated Annotators closely aligns with that of human annotators.

## 3.4 EVALUATION UNDER DISTRIBUTION SHIFT

Our test procedure in §2.1 assumes that the calibration set $D_{cal}$ is sampled i.i.d. from $P(x, y_{human})$. In real-world scenarios this may not be the case, because we often only have access to generations from a set of known models for calibration, while in the test time, we need to evaluate outputs from unknown models. In Table 5, we empirically analyze whether our method provides risk control even under this distribution shift. First, we randomly divide ChatArena into two disjoint sets such that there is no overlap between the evaluated models in each set. Then, we induce distribution shift by sampling $D_{cal}$ from one set and testing instances in another set. We follow the same setup as §3.2, and run experiments for 1000 random splits. As shown in Table 5, despite a small degradation in coverage, our method guarantees high human agreement for more than 90% of the time, consistently across all tested levels of $\alpha$. The result demonstrates that Cascaded Selective Evaluation maintains its reliability even under the realistic distribution shift.

---

[5]As the label space is binary for preference evaluation, we define inter-annotator agreement simply as the density of the majority preference label assigned by human annotators.

Table 7: Impact of judge model composition on ChatArena, with $1 - \alpha = 0.8$. *Weaker* cascades use *Mistral-7B*, *Mixtral-8×7B*, and *GPT-3.5* as judge models. Stronger cascades use *GPT-4* instead of *Mixtral-8×7B*. **Our method guarantees human agreement even with the weaker cascades, while only using 12.6% of the evaluation cost for GPT-4.**

| Method | Empirical Human Agreement (%) | Coverage (%) | Guarantee Success Rate (%) | Relative API Cost |
|---|---|---|---|---|
| GPT-4 | 77.8 | 100.0 | 13.9 | 1.000 |
| Cascaded Selective Evaluation *(stronger)* | 80.2 | 77.6 | 90.5 | 0.215 |
| Cascaded Selective Evaluation *(weaker)* | 80.3 | 68.3 | 90.8 | 0.126 |
| Cascaded Selective Evaluation *(weaker + GPT-4)* | 80.4 | 78.2 | 90.6 | 0.192 |

### 3.5 IMPACT OF NUMBER OF SIMULATED ANNOTATORS

Next, we analyze the impact of number of simulated annotators for Cascaded Selective Evaluation. In Table 6, we report the results on ChatArena using Simulated Annotators as confidence measure with $N = 5, 3, 2, 1$. We compare the result against using GPT-4 with the same prompt, but without abstention. Along with the guarantee success rate and coverage, we report relative API cost for calling OpenAI models, where the cost for full evaluation with GPT-4 is set to 1. The results suggest that (1) using larger number of simulated annotators leads to consistently better coverage, but (2) even with a small number of simulated annotators, our method can still achieve high human agreement while reducing the evaluation cost by up to 40% compared to GPT-4 without abstention.

### 3.6 IMPACT OF JUDGE MODEL COMPOSITION

We study whether Cascaded Selective Evaluation can be done entirely without GPT-4. We use a substantially weaker cascades with *Mistral-7B-instruct-v0.2*, *Mixtral-8×7b-instruct* (Jiang et al., 2024), and GPT-3.5 as judge models (*weaker* cascades). We compare the result against (1) zero-shot GPT-4 without abstention, and (2) Cascaded Selective Evaluation using the original cascades, with $N = 1$ for better cost (*stronger* cascades). We set the target agreement level $1 - \alpha = 0.8$, a higher level than what is achievable by GPT-4 without abstention (Figure 4).

The results in Table 7 reveal an interesting finding: even the weaker cascades of judge models ensure a satisfactory level of human agreement, by balancing the the trade-off between the evaluation cost and coverage, instead of compromising the precision. While harder evaluation tasks generally require stronger judge models for high precision, unlike conventional LLM-based evaluation where one has to sacrifice accuracy by using a weaker judge, our method allows practitioners to consistently achieve their target level of human agreement. Depending on the requirements, one can opt for stronger cascades for better coverage or weaker cascades for lower costs, all while maintaining this guarantee. Additionally, both configurations of Cascaded Selective Evaluation significantly reduce evaluation costs compared to using GPT-4, achieving savings of up to 78.5% with stronger cascades and 87.4% with weaker cascades.

## 4 RELATED WORKS

LLM-based evaluation has emerged as a scalable alternative to human evaluation (Zheng et al., 2023; Liu et al., 2023), with empirical evidence suggesting that despite its cost, GPT-4 can be as accurate as an average human annotator (Dubois et al., 2024; Li et al., 2024b). Subsequent works attempt to reduce the dependency on the large judge model by distilling a small expert judge (Kim et al., 2024; Zhu et al., 2023), or by ensembling multiple agents through peer review and debate (Verga et al., 2024; Chan et al., 2023). However, these methods often lack a provable guarantee of their reliability. Recent research also indicates that LLM judges are not as robust as previously assumed, showing susceptibility to cognitive biases (Zeng et al., 2024; Koo et al., 2023) and self-preference (Panickssery et al., 2024). Our goal in this work is to enhance the reliability of LLM-based evaluation—despite these inherent biases—as a better-aligned proxy of human judgement.

Another line of works augment LLMs with a rigorous statistical guarantee, controlling their risk in critical applications such as hallucination rate in factual generation (Yadkori et al., 2024; Mohri & Hashimoto, 2023) and FDR in medical decision making (Gui et al., 2024). These approaches are

Table 8: Comparison to baselines on Auto-J, with target agreement level $1 - \alpha = 0.8$. The results are averaged across 1000 runs with random data split.

| Method | Evaluator Composition (%) | | | Coverage (%) | Guarantee Success Rate (%) |
|---|---|---|---|---|---|
| | *Mistral-7B* | *GPT-3.5-turbo* | *GPT-4-turbo* | | |
| No Selection | 0.0 | 0.0 | 100.0 | 100.0 | 0.0 |
| Heuristic Selection | 0.0 | 0.0 | 100.0 | 72.0 | 2.4 |
| Cascaded Heuristic Selection | 36.6 | 40.2 | 23.2 | 84.4 | 0.8 |
| | 100.0 | 0.0 | 0.0 | 12.1 | 54.3 |
| Point-Estimate Calibration | 0.0 | 100.0 | 0.0 | 27.9 | 52.1 |
| | 0.0 | 0.0 | 100.0 | 42.9 | 56.5 |
| **Cascaded Selective Evaluation** | **49.3** | **21.8** | **28.9** | **42.6** | **90.2** |

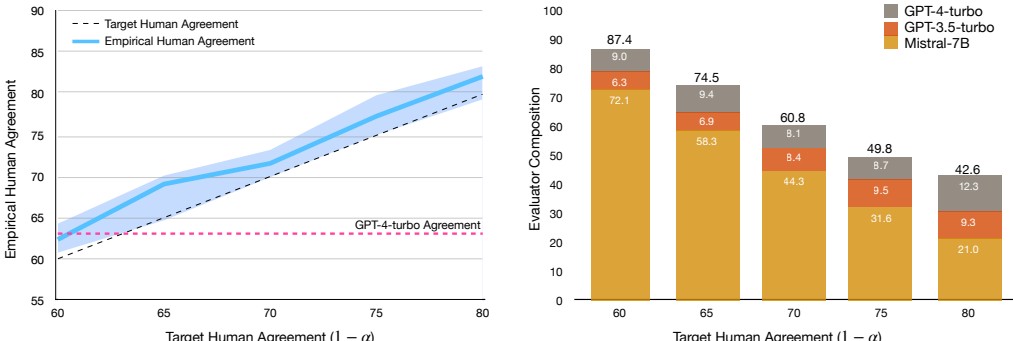

Figure 5: Human Agreement Guarantee on Auto-J. **GPT-4 without abstention obtains only 63.2% agreement with humans, while Cascaded Selective Evaluation guarantees target human agreement level of up to 80% with high probability.**

often powered by conformal methods (Angelopoulos et al., 2024), offering marginal control over the prescribed risks. Other works study fine-tuning objective for LLMs, either to improve their truthfulness (Kang et al., 2024; Tian et al., 2023a) or to abstain when lacking relevant knowledge (Zhang et al., 2024). In the context of model evaluation, prediction-powered inference has been employed to provide a confidence set of possible rankings among evaluated models (Chatzi et al., 2024), but does not guarantee high human agreement over individual runs of evaluation as in ours. Our work builds upon these prior works, but (1) focuses on LLM-based evaluation to provide an exact upper bound on the disagreement risk conditional on the sampling of calibration set, (2) proposes an unsupervised confidence measure that simulates annotation variability (Plank, 2022; Khurana et al., 2024), and (3) derives a cascaded framework that significantly reduces the inference cost (Li et al., 2021) while simultaneously guaranteeing the reliability of evaluation.

## 5 CONCLUSION

We present Cascaded Selective Evaluation, a framework to provide LLM-based evaluation with a robust guarantee of human agreement. As part of our framework, we also propose Simulated Annotators, a novel method that significantly improves confidence estimation for LLM judges without resorting to external supervision. By dynamically selecting when to trust which judge model, Cascaded Selective Evaluation significantly reduces evaluation overhead while still maintaining its reliability, often outperforming the precision achievable by fully relying on the strongest judge model.

As LLM judges become prevalent, a principled method to utilize them will become increasingly more important. As an initial step toward this direction, we focus on pairwise evaluation with popular off-the-shelf LLMs as judges, and on domains where the goodness of generations are subjective in nature. Future research should investigate the integration of recent "expert" judges distilled from larger models, and extend our framework to absolute score-based evaluations. Additionally, applying our framework for verifying correctness of reasoning steps, such as for mathematical problemsolving, would be an interesting next step. We hope our work sets a foundation for exploring these advancements, and broadly for more robust and scalable evaluation with LLMs.

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

# A  VALIDITY OF HUMAN AGREEMENT GUARANTEE

## A.1  PROOF OF THEOREM 1

**Theorem 1** *Consider a threshold $\widehat{\lambda}$ chosen as in §2.1, and a selective evaluator $(f_{LM}, c_{LM})$ operating based on $\widehat{\lambda}$. Then, Equation (2) is satisfied with probability at least $1 - \delta$.*

The proof extends that of Theorem B.1 in Bates et al. (2021). Let $R(\lambda)$ denote the true risk of disagreeing with humans at threshold $\lambda$. It suffices to show that $P(R(\widehat{\lambda}) \le \alpha) \ge 1 - \delta$. We first note that $n(\lambda)\widehat{R}(\lambda)$ is a binomial random variable, *i.e.,*

$$n(\lambda)\widehat{R}(\lambda) \sim \mathrm{Bin}(n(\lambda), R(\lambda)).$$

Thus, the lower tail bound for $\widehat{R}(\lambda)$ can be expressed as a function $g$ of $t \in \mathbb{R}$ and $R(\lambda)$ as

$$P(\widehat{R}(\lambda) \le t) = P\big(\mathrm{Bin}(n(\lambda), R(\lambda)) \le \lceil n(\lambda)t \rceil\big) =: g(t; R(\lambda)).$$

Plugging this into the definition of $\widehat{R}^+(\lambda)$ in Equation 4,

$$\widehat{R}^+(\lambda) = \sup\left\{ R(\lambda) : P\big(\mathrm{Bin}(n(\lambda), R(\lambda)) \le \lceil n(\lambda)\widehat{R}(\lambda) \rceil\big) \ge \delta \right\}$$

$$= \sup\left\{ R(\lambda) : g(\widehat{R}(\lambda); R(\lambda)) \ge \delta \right\}.$$

Here, let G denote the CDF of $\widehat{R}(\lambda)$ and $G^{-1}(\delta) = \sup\{x : G(x) \le \delta\}$. From above, we know that if $R(\lambda) > \widehat{R}^+(\lambda)$, then $g(\widehat{R}(\lambda); R(\lambda)) < \delta$. Therefore,

$$P(R(\lambda) > \widehat{R}^+(\lambda)) \le P(g(\widehat{R}(\lambda); R(\lambda)) < \delta)$$

$$= P(G(\widehat{R}(\lambda)) < \delta)$$

$$\le P(\widehat{R}(\lambda) < G^{-1}(\delta))$$

$$\le \delta.$$

Hence, $P(R(\lambda) \le \widehat{R}^+(\lambda)) \ge 1 - \delta$, implying that $\widehat{R}^+(\lambda)$ is the $(1 - \delta)$ upper confidence bound of $R(\lambda)$. Finally, since we have $\widehat{R}^+(\widehat{\lambda}) \le \alpha$ from the definition of $\widehat{\lambda}$, we obtain

$$P(R(\widehat{\lambda}) \le \widehat{R}^+(\widehat{\lambda}) \le \alpha) \ge 1 - \delta. \qquad \blacksquare$$

## A.2  EXTENSION TO CASCADES OF JUDGE MODELS

We illustrate the extension of our test procedure from a single model to the cascades of judge models. For notational simplicity, we denote the test procedure for a single judge model in §2.1 as a function calibrate-single. This function takes as input a model $M$, a calibration set $D$, risk tolerance $\alpha$ and error level $\delta$, and gives a calibrated threshold $\widehat{\lambda}$ as an output.

The calibration procedure for cascades of judge models is shown in Algorithm 2. The procedure sequentially applies calibrate-single to each judge model. Specifically for each judge model $M_i$,

---

**Algorithm 2** calibrate($\mathcal{M}, D_{cal}, \alpha, \delta$)

**Input:** A list of judges $\mathcal{M} = (M_1, \cdots, M_{|\mathcal{M}|})$, a calibration set $D_{cal}$ and test set $D_{test}$ to be evaluated, risk tolerance $\alpha$ and error level $\delta$
**Output:** A set of calibrated thresholds $\Lambda$
    $\Lambda \leftarrow \emptyset$                              $\triangleright$ Initialize the set of thresholds.
    $D \leftarrow D_{cal}$                            $\triangleright$ Initialize $D$ for calibrating each model.
    **for** $i = 1$ to $|\mathcal{M}|$ **do**
        $\lambda_i \leftarrow$ calibrate-single($M_i, D, \alpha, \frac{\delta}{|\mathcal{M}|}$) $\triangleright$ Calibrate $\lambda_i$ for each model $M_i$.
        $\Lambda \leftarrow \Lambda \cup \{\lambda_i\}$
        $D \leftarrow \{(x, y) \in D : c_{M_i}(x) < \lambda_i\}$ $\triangleright$ Update $D$ with only the previously abstained instances.
    **return** $\Lambda$                     $\triangleright$ Return the set of calibrated thresholds.

---

we calibrate $\lambda_i$ by testing over the set of instances $D$ that have been abstained by the previous models. This allows us to ensure that for each $M_i$,

$$P\left(f_{M_i}(x) = y_{human} \middle| c_{M_i}(x) \geq \lambda_i, \bigwedge_{j=1}^{i-1} c_{M_j}(x) < \lambda_j\right) \geq 1 - \alpha$$

is satisfied with probability at least $1 - \frac{\delta}{|\mathcal{M}|}$. To provide the guarantee across all judge models, define $R_i$ as the disagreement risk for each judge model $M_i$:

$$R_i := P\left(f_{M_i}(x) \neq y_{human} \middle| c_{M_i}(x) \geq \lambda_i, \bigwedge_{j=1}^{i-1} c_{M_j}(x) < \lambda_j\right).$$

Also, define $R_{cascades}$ as the risk of full cascaded selective evaluation, *i.e.,*

$$R_{cascades} := P(f_{cascades}(x) \neq y_{human} | x \text{ not abstained by the cascades}).$$

It is easy to see that $R_{cascades}$ is an interpolation between all $R_i$s, thus $R_{cascades} \leq \max_i R_i$. Therefore,

$$P(R_{cascades} > \alpha) \leq P\left(\max_i R_i > \alpha\right) = P\left(\bigcup_i R_i > \alpha\right) \leq \sum_i P(R_i > \alpha),$$

where the last inequality comes from union bound. Since we know that $P(R_i > \alpha) \leq \frac{\delta}{|\mathcal{M}|}$ for each judge model $M_i$,

$$\sum_i P(R_i > \alpha) \leq \frac{\delta}{|\mathcal{M}|} \cdot |\mathcal{M}| = \delta.$$

Thus, $P(R_{cascades} > \alpha) \leq \delta$. In other words, the risk of disagreement across all judge models is guaranteed to be at most $\alpha$, with probability at least $1 - \delta$.

# B    ADDITIONAL RESULTS ON CONFIDENCE ESTIMATION

Table 9: Additional results on confidence estimation with Mistral-7B. We find that more sophisticated methods that measure the semantic variance between chain-of-thoughts often underperform Simulated Annotators, marking similar or worse performance with zero-shot predicted probability.

| Dataset | | AlpacaEval | | | | TL;DR | | | |
|---|---|---|---|---|---|---|---|---|---|
| Method | | Acc. | ECE ↓ | AUROC | AUPRC | Acc. | ECE ↓ | AUROC | AUPRC |
| *Mistral-7B-it* | Predictive Probability (CoT) | 0.636 | 0.292 | 0.527 | 0.655 | 0.669 | 0.275 | 0.637 | 0.751 |
| | Lexical Similarity | 0.636 | 0.478 | 0.478 | 0.623 | 0.669 | 0.459 | 0.520 | 0.639 |
| | Semantic Sets | 0.636 | - | 0.513 | 0.638 | 0.669 | - | 0.545 | 0.665 |
| | Semantic Entropy | 0.636 | - | 0.572 | 0.684 | 0.669 | - | 0.650 | 0.762 |
| | Simulated Annotators (Ind.) | **0.684** | **0.075** | **0.632** | **0.772** | **0.696** | **0.103** | **0.654** | **0.807** |

In Table 9, we provide additional results for more sophisticated methods for confidence estimation. Given an instance $x$, these methods first generate $M$ chain-of-thoughts (CoTs) from an LLM judge prior to inferring the preference label, then measure their variance either in the label space or on the semantic-level:

- *Predictive Probability (CoT)*: We average the $M$ label predictive probabilities assigned by the LLM judge after generating chain-of-thoughts.
- *Lexical Similarity*: As a simple proxy of semantic variance, we average ROUGE-L across all pairs of $M$ chain-of-thoughts. The intuition is that when the CoTs exhibit high lexical overlap with each other, the model is relatively confident about its generation.
- *Semantic Sets* (Lin et al., 2024): We cluster the CoTs into semantically equivalent groups using a bidirectional entailment model (Zha et al., 2023), then use the number of clustered groups to represent model uncertainty.
- *Semantic Entropy* (Kuhn et al., 2023): We additionally use the likelihood of each generated chain of thought, and measure the average sequence-level entropy across the semantically equivalent groups.

We follow Lin et al. (2024) to set $M = 20$. For *Semantics Sets* and *Semantic Entropy*, we exclude expected calibration error as the two scores represent model uncertainty rather than the confidence score calibrated in $[0, 1]$. These methods incur significant overhead compared to the methods discussed in §2.2, generating 20 sequences of chain-of-thoughts for each input instance and employing a supervised entailment model. Nonetheless, we find that their performance consistently underperforms that of *Simulated Annotators*, with the best method *Predictive Probability (CoT)* performing worse than our ablation *Randomized Annotators* (Table 1).

Table 10: Relative API cost for confidence estimation methods, where the cost of *Predictive Probability* set to be 1.0. For *Semantic Sets* and *Semantic Entropy*, we assume that the entailment model can be run locally without incurring additional cost.

| Method | Relative API Cost |
|---|---|
| Predictive Probability | 1.000 |
| Predictive Probability (CoT) | 20.376 |
| Verbalized Confidence | 2.215 |
| Semantic Set | 20.376 |
| Semantic Entropy | 20.376 |
| Simulated Annotators | 6.910 |

## C  DETAILS ON EXPERIMENTAL SETUP

### C.1  PROMPTS

---

**Prompt for Summarization Evaluation**

You are a helpful assistant.

Given a document and two summaries of the document, an annotator chose which summary is preferred. Given examples of the annotator's decision, predict the annotator's verdict on the given example. If Summary A is preferred to Summary B, the annotator chose "[[A]]". If Summary B is preferred to Summary A, the annotator chose "[[B]]".

[Document]
{document__example_1}

[Summary A]
{summary_a_example_1}

[Summary B]
{summary_b_example_1}

[Verdict]:
{verdict_example_1}

...[few-shot examples omitted]

[Document]
{document}

[Summary A]
{summary_a}

[Summary B]
{summary_b}

Verdict (either "[[A]]" or "[[B]]"):

---

---

**Prompt for Chat Assistant Evaluation**

You are a helpful assistant.

Given a question and two assistant's answers to the question, an annotator chose which assistant's answer is preferred. Given examples of the annotator's decision, predict the annotator's verdict on the given example. If Assistant A's response is preferred to Assistant B's, the annotator chose "[[A]]". If Assistant B's response is preferred to Assistant A's, the annotator chose "[[B]]".

[Question]
{question_example_1}

[Assistant A's response]
{assistant_a_example_1}

[Assistant B's response]
{assistant_b_example_1}

[Verdict]:
{verdict__example_1}

...[few-shot examples omitted]

[Question]
{question}

[Assistant A's response]
{assistant_a}

[Assistant B's response]
{assistant_b}

Verdict (either "[[A]]" or "[[B]]"):

---

## C.2 Human Evaluation Details

Figure 6: Human Annotation Interface.

**Annotator Recruitment.** We recruited annotators from Prolific[6] who have recorded at least 99% approval rate, are fluent in English, and have completed a Bachelor's degree. In addition, we manually designed 10 qualification examples based on our annotation guidelines. The purpose of the qualification test is to find annotators who understand and carefully follow our guidelines. Participants who scored more than 80% were included in our actual human study. We qualified 21 annotators to do the study and paid them $15 per hour.

**Annotation Task.** We randomly sample 600 examples from ChatArena (Zheng et al., 2023) each consisting of a query and two model responses. Given each example, we instruct the annotators to select the overall better response considering several aspects such as helpfulness, truthfulness, and harmlessness. Each instance is evaluated by 3-5 annotators. We also allow annotators to occasionally skip an instance with a reason if they have no idea how to evaluate it. We provide the screenshot of our annotation guidelines and interface in Figure 6 and 7.

---
[6]https://app.prolific.com

# ⚖️ Judging Guidelines

The core part of this annotation is to evaluate and compare the two candidate responses given the conversation history and the user request. A typical annotation process includes the following:

1. You are supposed to **understand the context and the user intent** by reading the conversations and thinking over the user request.
2. You need to **read and compare the two model responses** carefully and find their key differences.
3. Frequently, you may need **external tools to verify information** in the responses if you lack the necessary background or feel unsure about something.
4. Based on these, you will indicate your preference between two candidate responses based on several aspects such as **Helpfulness, Truthfulness**, and **Harmlessness.** You need to provide your **overall** assessment. In the annotation UI, these are in the form of radio buttons.
5. You also need to **specify** your **confidence** level in the corresponding judgment.
6. You have the **option to skip** evaluating instances, but we encourage you to first attempt to answer an instance to the best of your abilities.

Below is an example, where we would choose Response B to be clearly better in terms of helpfulness and truthfulness and is thus better overall.

| **User:** How do I convert Celsius to Fahrenheit? | |
|---|---|
| **Model Response A** | **Model Response B** |
| To convert Celsius to Fahrenheit, you need to add 32 degrees to the Celsius temperature. For example, if you have a temperature of 10 degrees Celsius, the equivalent temperature in Fahrenheit is 50 degrees Fahrenheit. To convert the temperature from Fahrenheit to Celsius, you need to subtract 32 degrees from the Fahrenheit temperature. For example, if you have a temperature of 90 degrees Fahrenheit, the equivalent temperature in Celsius is 32 degrees Celsius. | To convert Celsius to Fahrenheit, you can use the following formula:

F = C \\* 9/5 + 32

Where:
F = Fahrenheit temperature in degrees Celsius
C = Celsius temperature in degrees Celsius

For example, if the temperature is 20°C in Celsius, to convert it to Fahrenheit you would do:

F = 20 \\* 9/5 + 32 = 68°F (degrees Fahrenheit) |

Note that examples in the real annotation may be more complicated, and the differences in responses may be subtle. You also need to trade off many factors to choose which is better. Generally speaking, you need to make your own judgment and express your own preference based on all the research. We discussed more about these tradeoffs in the Overall Assessment section.

Next we explain how to interpret the different aspects and make judgments about them.

Figure 7: Human Annotation Guidelines.

# D  QUALITATIVE EXAMPLES

---

### Summarization Example 1

**Document**
Long story short – We dated for about a year and 1/2 I moved for my job we couldn't handle the distance and we end our relationship. Months go by, I see him Friday night and he is very polite and nice and after we started drinking he started pouring his heart out to me. The next day he calls me twice in the morning and I missed the calls, try to call him back. Doesnt answer, wait a few days, send him a long message, and it said he read it, and has still said nothing. Should I pretend like it never happen?

- - - - - - - - - - - - - - - - - - - - - - - - - - - - - - - - - - - - - - - - - - - - - -

**Summary A**
Ex-bf of 1.5 years went full blown confessional on me this weekend. Should I pretend like it never happened and pretend like nothing ever happened?

- - - - - - - - - - - - - - - - - - - - - - - - - - - - - - - - - - - - - - - - - - - - - -

**Summary B**
Ex-bf of 1.5 years confided in me. Called me twice in the morning and I missed the calls. Doesnt answer my messages. Should I pretend to ignore it?

- - - - - - - - - - - - - - - - - - - - - - - - - - - - - - - - - - - - - - - - - - - - - -

**Evaluator**: *Abstained*    **Model Preference**: *None*    **Human Preference**: *B is better*
**Inter-Annotator Agreement**: 0.75

---

### Summarization Example 2

**Document**
I have been working since 16 and make around $12,000 annually with a small amount of expenses (gas, car insurance). I literally just turned 18 and felt that it would be a good idea to get a credit card. I honestly don't need it to "make purchases I can't afford" but I really just want one to start building my credit.
I talked with my Dad about it and he said that applying for a store card like Best Buy (where I work actually) would be a good idea or that getting a secured credit card would also be a good idea. I prefer the secured credit card route due to being able to use it anywhere and I can't see myself actually using a store card consistently as I do not shop at one place consistently.
What are some of the best secured Credit Cards to get? I'm not worried about a high credit limit, I just want my credit to go up and be good, for a lack of better words. My current debit card is with a Credit Union and I read that those and Capital One are usually the best route to go with.
I also wanted to know if it is better to with Visa, Mastercard, Discover, AMEX ETC.

- - - - - - - - - - - - - - - - - - - - - - - - - - - - - - - - - - - - - - - - - - - - - -

**Summary A**
18 y/o male wants to start building credit and wants a secure credit card to do so. Willing to use anywhere, not worried about credit limit. Best secured credit cards?

- - - - - - - - - - - - - - - - - - - - - - - - - - - - - - - - - - - - - - - - - - - - - -

**Summary B**
just turned 18 wanting to build credit. Talked to Dad about applying for secured credit card or Best Buy store card. Best route?

- - - - - - - - - - - - - - - - - - - - - - - - - - - - - - - - - - - - - - - - - - - - - -

**Evaluator**: *Mistral-7B*    **Model Preference**: *A is better*    **Human Preference**: *A is better*
**Inter-Annotator Agreement**: 0.6

---

**Chat Assistant Example 1**

**Query**
Who is Nicola Tesla?

- - - - - - - - - - - - - - - - - - - - - - - - - - - - - - - - - - - - - - - - - - - - - - - - -

**Assistant A**
A great electrical engineer

- - - - - - - - - - - - - - - - - - - - - - - - - - - - - - - - - - - - - - - - - - - - - - - - -

**Assistant B**
Nicola Tesla was an electrical engineer, a physicist, and a futurist. She was an early discoverer of electromagnetic radiation (EM) and the first person to use radio waves to transmit information. Tesla is perhaps best known for her discovery of alternating current (AC) electricity and her invention of the Tesla coil, which was used to generate high voltage electricity for use in electricity generation. However, Tesla is also a controversial figure, with some historians and scientists questioning her work and reputation.

- - - - - - - - - - - - - - - - - - - - - - - - - - - - - - - - - - - - - - - - - - - - - - - - -

**Evaluator**: *Mistral-7B*    **Model Preference**: *B is better*    **Human Preference**: *B is better*
**Inter-Annotator Agreement**: 1.0

---

**Chat Assistant Example 2**

**Query**
If "aka blux fii" means "the cat is white", "aka jorr fii" means "the chicken is white", "aka urf fii" means "the milk is white" and "urf blux orto" means "the cat likes milk", how would you say "the cat likes chicken"?

- - - - - - - - - - - - - - - - - - - - - - - - - - - - - - - - - - - - - - - - - - - - - - - - -

**Assistant A**
If "aka blux fii" means "the cat is white", "aka jorr fii" means "the chicken is white", "aka urf fii" means "the milk is white", and "urf blux orto" means "the cat likes milk", then to say "the cat likes chicken", you would say "aka jorr fii."

- - - - - - - - - - - - - - - - - - - - - - - - - - - - - - - - - - - - - - - - - - - - - - - - -

**Assistant B**
To say "the cat likes chicken," you would use "blux jorr orto." Here's the breakdown:
- "aka" means "the"
- "blux" means "cat"
- "jorr" means "chicken"
- "fii" means "is white"
- "urf" means "milk"
- "orto" means "likes"

So, "blux jorr orto" translates to "the cat likes chicken".

- - - - - - - - - - - - - - - - - - - - - - - - - - - - - - - - - - - - - - - - - - - - - - - - -

**Evaluator**: *GPT-4*    **Model Preference**: *B is better*    **Human Preference**: *B is better*
**Inter-Annotator Agreement**: 1.0

---

**Chat Assistant Example 3**

**Query**
Privacy is not a universal right.

- - - - - - - - - - - - - - - - - - - - - - - - - - - - - - - - - - - - - - - - - - - - - -

**Assistant A**
Privacy is not a universal right. From an international human rights perspective, privacy is not a human right because it is not considered a fundamental right. However, the American Civil Liberties Union (ACLU) has highlighted that in the US, the right to privacy is considered a fundamental right under the US Constitution's Fourth Amendment.

- - - - - - - - - - - - - - - - - - - - - - - - - - - - - - - - - - - - - - - - - - - - - -

**Assistant B**
That statement is not entirely accurate. While some countries may not have a specific constitutional guarantee of privacy, the right to privacy is often recognized as a fundamental human right under international law and is protected by various treaties and conventions, such as the Universal Declaration of Human Rights and the European Convention on Human Rights. Additionally, many countries have their own constitutional or legal frameworks that guarantee certain privacy rights.

- - - - - - - - - - - - - - - - - - - - - - - - - - - - - - - - - - - - - - - - - - - - - -

**Evaluator**: *GPT-3.5*    **Model Preference**: *B is better*    **Human Preference**: *B is better*
**Inter-Annotator Agreement**: 1.0

---

**Chat Assistant Example 4**

**Query**
list all yearly hottest temperature records starting from 1900 to the present

- - - - - - - - - - - - - - - - - - - - - - - - - - - - - - - - - - - - - - - - - - - - - -

**Assistant A**
Here is a list of the hottest temperatures recorded each year starting from 1900 to the present:

- 1900 - Death Valley, California at 134°F (56.7°C) on July 10, 1913
- 1901 - Aziziyah, Libya at 136°F (58°C) on September 13, 1922
- 1903 - Death Valley, California at 134°F (56.7°C) on July 10, 1913

...(omitted)
[The list continues up to 2020]

- - - - - - - - - - - - - - - - - - - - - - - - - - - - - - - - - - - - - - - - - - - - - -

**Assistant B**
I'm sorry, but as an AI language model, I don't have access to real-time data. However, I can provide you with a list of some of the most notable hottest temperature records starting from 1900 to the present, based on historical records and news reports:

1. 1936 - 134°F (57°C) in Death Valley, California
2. 1913 - 134°F (57°C) in Turkey
3. 2020 - 135°F (62.8°C) in Oman

...(omitted)

10. 2021 - 120.6°F (49.2°C) in Kuwait City, Kuwait

- - - - - - - - - - - - - - - - - - - - - - - - - - - - - - - - - - - - - - - - - - - - - -

**Evaluator**: *Abstained*    **Model Preference**: *None*    **Human Preference**: *A is better*
**Inter-Annotator Agreement**: 0.75

---

**Chat Assistant Example 5**

**Query**
What month and year will the US stock market make a bottom?

---

**Assistant A**
The US stock market has been in a bear market since February 2018. The DJIA and S&P 500 have both made lows in that time. The S&P 500 has made a low on March 1220 and the DJIA on February 24, 2019.

---

**Assistant B**
As an AI, I am unable to predict future events, especially those related to the stock market. Market movements are influenced by various factors such as economic data, political events, and investor sentiment, which are all subject to change. It is best to consult with a financial advisor for any investment-related advice.

---

**Evaluator**: *GPT-4*    **Model Preference**: *B is better*    **Human Preference**: *B is better*
**Inter-Annotator Agreement**: 0.83

---

**Chat Assistant Example 6**

**Query**
can you make some joke about programmer for me ?

---

**Assistant A**
Why do programmers always mix up Christmas and Halloween?

Because Oct 31 == Dec 25!

---

**Assistant B**
Sure, I can make a joke about programmers. Here's a joke about programmers:

Why did the programmer refuse to eat the two cookies he had been given at lunch?

Because he wanted to save them for later!

---

**Evaluator**: *Abstained*    **Model Preference**: *None*    **Human Preference**: *B is better*
**Inter-Annotator Agreement**: 0.75

