# OpenReview forum: "Trust or Escalate: LLM Judges with Provable Guarantees for Human Agreement"
_ICLR.cc/2025/Conference — ICLR 2025 Oral_

### Official Review · Reviewer_mWkk · 2024-10-30

**Soundness:** 3
**Presentation:** 3
**Contribution:** 3
**Rating:** 6
**Confidence:** 3

**Summary:**

The authors introduce a principled method for evaluating language models with guaranteed human agreement. They argue that evaluation should not solely rely on model preferences in pairwise comparisons. Instead, it should assess judge models' confidence and selectively decide when to trust their judgment. This selective framework ensures that the evaluation aligns with human agreement to a specified level. They introduce Simulated Annotators, a new method for confidence estimation that enhances judge calibration, enabling comprehensive evaluation. Their Cascaded Selective Evaluation uses cost-effective models initially, escalating to stronger ones as needed, maintaining human agreement guarantees. Results show this method achieves over 80% human agreement.

**Strengths:**

1. Robust Human Agreement: The Cascaded Selective Evaluation framework offers a strong guarantee of alignment with human judgment, enhancing the reliability of LLM-based evaluations.

2. Improved Confidence Estimation: The introduction of Simulated Annotators enhances the confidence estimation for LLM judges, allowing for more accurate evaluations without the need for external supervision.

3. Dynamic Judge Selection: By dynamically selecting which judge model to trust, the framework reduces evaluation overhead while maintaining reliability, often surpassing the precision of relying solely on the strongest judge model.

**Weaknesses:**

It is more like a theoretical work. Given that the experiments have proved effective in evaluating different tasks, I suggest the authors invest some effort in releasing a user-friendly toolkit based on this work, for other researchers to use.

**Questions:**

NA

---

> ### Author Response · Authors · 2024-11-18
>
> We thank the reviewer for the positive comments! We are glad that you view our work to enhance the reliability of LLM-based evaluations.
>
> &nbsp;
>
> ### Is this a theoretical work?
> While our suggested framework is grounded on the theories of inferential statistics, we believe our core contribution lies on its significant empirical results, that make LLM judges to be more reliable and (simultaneously) more cost-effective. We are happy to see that the reviewer also recognizes the framework’s effectiveness in improving LLM judges across different tasks.
>
> &nbsp;
>
> ### User-friendly toolkit
> We appreciate this suggestion! We are already working on implementing a high-level interface of our method where users only have to put in the name of judges and target agreement level. The framework then would automatically (1) calibrate the evaluation confidence, and (2) decide the abstention policy to provide the human agreement guarantee.
>
> We hope this addresses your comments!

---

> > ### Comment · Reviewer_mWkk · 2024-11-21
> >
> > Thanks for the response. I hope the community will have this toolkit soon!

---

> ### Author Response · Authors · 2024-11-25
>
> Dear reviewer mWkk,
>
> We have updated the submission and addressed the feedback from each reviewer. We are checking in to see if we have adequately addressed your concerns and whether you would consider adjusting the paper's rating.
>
> Thank you,

---

### Official Review · Reviewer_4Yrv · 2024-11-01

**Soundness:** 3
**Presentation:** 3
**Contribution:** 3
**Rating:** 8
**Confidence:** 5

**Summary:**

The authors propose a framework for LLM evaluation that includes a human agreement guarantee, that is called Cascaded Selective Evaluation. The evaluation task (in this case a pairwise evaluation) is given to a weak LLM judge and escalated to a stronger LLM judge, if the confidence level for the task does not exceed a certain threshold. As a confidence estimation method they propose Simulated Annotators, where multiple LLMs are used to simulate annotator preferences.

**Strengths:**

The paper is clearly structured and well written. The approach clearly improves the evaluation process supported by LLMs, while being cost effective by only considering stronger models (such as GPT-4-turbo) as a last escalation step. The authors provide meaningful experiments and show how the method can be extended in the future.

**Weaknesses:**

A weakness of the approach (for now), is that it has only been tested and applied in context of the pairwise preference evaluation. Nevertheless, the Simulated Annotators and Cascaded Selected Evaluation methods are promising. It will be interesting to see how the approach performs in other evaluation settings in the future. The question of when the models abstain could be answered in more detail. For now, it is only answered quantitatively, but not really qualitatively. In order to find out whether the examples the models are not sure about are the same ones about which the human judges also disagree, the examples need to be examined in more detail, i.e. more on the level of content. It would be interesting to see the ratio of examples where both humans and the models abstained from evaluation and to analyze some examples on a content level.

Small errors:
- Line 462: the the tradeoff

**Questions:**

- You analyzed the abstention rate quantitatively, but did you also do a qualitative analysis? Do the models abstain on the same type of pairs?

---

> ### Author Response · Authors · 2024-11-18
>
> Thank you for the positive endorsement of our work. We are happy that you found our work to be well-written and structured, our experiments to be meaningful, and overall to clearly improve LLM-based evaluation!
>
> &nbsp;
>
> ### On future extension of Cascaded Selective Evaluation
> As noted in the conclusion, we focused on pairwise evaluation as it represents the most widely used applications of LLM judges, but in principle, our framework can be generalized beyond the pairwise format. We are excited to work on the future extension of Cascaded Selective Evaluation for diverse evaluation scenarios, such as Likert Scale.
>
> &nbsp;
>
> ### Qualitative analysis on abstained examples
> We appreciate this suggestion. We followed your comments to analyze the abstained examples in a more qualitative perspective, by (1) categorizing prompts based on their contents and identifying characteristics of abstention policy, (2) evaluating whether humans would have “abstained” for the same samples with the LLM judges. We’ve included 8 qualitative examples across summarization evaluation / chat assistant evaluation **(Appendix C)**, which would help making it clearer the insights from the qualitative analysees. We will make a clearer pointer to these examples in the main section.
>
>
> ***Content-based categorization*** \
> We first categorized prompts based on their contents and checked if there’s distinctive characteristics of model abstention policy. We followed [1] to divide prompts in ChatArena into the following 8 types - Advice, Communication (ChitChat), Writing, Reasoning, Knowledge-Aware, Unsafe Query, NLP Tasks, and Others, and used GPT-4o to automatically categorize each prompt into one of the 8 types. Then, we measured (1) the average human inter-annotator agreement, (2) ratio of machine-abstained samples, (3) the ratio of samples evaluated by each judge model in the cascades per each category.
>
> |                               | Advice | Communication (chitchat) | Writing | Reasoning | Knowledge Aware | Unsafe Query | NLP Tasks | Others |
> |:-----------------------------:|:------:|:------------------------:|:-------:|:---------:|:---------------:|:------------:|:---------:|:------:|
> |           Human IAA           |  0.78  |           0.76           |   0.78  |    0.85   |       0.76      |     0.92     |    0.78   |  0.81  |
> |    Machine Abstention Rate    |  0.52  |           0.51           |   0.53  |    0.50   |       0.55      |     0.24     |    0.60   |  0.46  |
> |   Mistral-7B Evaluation Rate  |  0.18  |           0.19           |   0.08  |    0.15   |       0.10      |     0.56     |    0.15   |  0.20  |
> | GPT-3.5-turbo Evaluation Rate |  0.24  |           0.22           |   0.23  |    0.15   |       0.20      |     0.16     |    0.15   |  0.24  |
> |  GPT-4-turbo Evaluation Rate  |  0.06  |           0.08           |   0.16  |    0.20   |       0.15      |     0.04     |    0.10   |  0.10  |
>
> The results are as shown above. While there is no salient trend across the categories, one notable outlier in machine abstention rate is in Unsafe Query, with significantly lower abstention rate (0.24) than others. We posit that the instruction-tuning process that incorporates safety training could have led the LLM judges to be confident in their evaluation for choosing a safer choice for unsafe query. An example of this category can be found in Chat Assistant Example 3 in the appendix.
>
> Meanwhile, we find that GPT-4-turbo tends to be more involved in reasoning type of prompts than e.g. ChitChat, which follows our intuition that harder tasks should require stronger judges in general (an example is provided in **Chat Assistant Example 2 in Appendix C**). Overall, we generally find that the machine-abstained examples tend to require subjective evaluation in nature, e.g. asking for a preference between two jokes **(Chat Assistant Example 6 in Appendix C)**.
>
> [1] Julong Li et al. Dissecting Human and LLM Preferences. 2024.

---

> > ### Author Response · Authors · 2024-11-18
> >
> > ***Analyzing overlap between human abstention and model abstain*** \
> > Since there was no abstention option given for human annotators, we do not have a direct access to what types of samples humans would have abstained from evaluating. However, as a proxy of human abstention, we may consider those samples that resulted in low inter-annotator agreement of 0.5~0.6 (i.e., nearly half of the annotators disagree with the other half). The model abstention rate in these samples, against the abstention rate on samples with IAA=1.0 (i.e. perfect agreement) is presented below:
> >
> > |       Cases       | Machine Abstention Rate |
> > |:-----------------:|:-----------------------:|
> > | Human "Abstained" |          0.803          |
> > | Human "Confident" |          0.314          |
> >
> > Consistently with the results in **Table 4**, when humans tend to disagree with each other, LLM judges tend to abstain from evaluation; when humans agree with each other, machines tend to be more confident as well. In other words, human abstention tends to correlate well with machine abstention policy.

---

> > > ### Comment · Reviewer_4Yrv · 2024-11-19
> > > **Response**
> > >
> > > Thank you very much for your detailed response and for considering my suggestions and giving some additional very interesting insights into your method.
> > > As I was already convinced, that your paper is very good and should be accepted at the conference, my rating remains unchanged.

---

> > > > ### Author Response · Authors · 2024-11-25
> > > >
> > > > Thank you again for the positive and thoughtful comments!

---

### Official Review · Reviewer_ZdoK · 2024-11-03

**Soundness:** 3
**Presentation:** 3
**Contribution:** 3
**Rating:** 8
**Confidence:** 3

**Summary:**

Current NLP paradigms have shifted to using LLMs as judges for annotations instead of humans. This paper asks how we can guarantee the reliability of using these LLMs for evaluation through human agreement.

Given an input, how likely would humans agree with the model judgment?

For this, the approach is to use selective evaluation: the model can abstain if not confident enough to make a prediction.

The conventional way of using the predictive probability as confidence estimates doesn't show to be reliable so two methods are proposed:
1. Simulated annotators: simulate annotator preferences through In-Context Learning and the confidence is estimated through the agreement ratio.
2. Cascaded Selective Evaluation: Start with a cheaper model and go to stronger models when the previous model is less confident.

This reduces the evaluation overhead with high agreement guarantees. Their abstention policy aligns with human subjectivity.

**Strengths:**

1. The paper tackles an important issue, we need more human subjectivity at the forefront of LLM evaluation.
2. In essence, the Simulated Annotators technique is ensembling, so using a simple and popular technique as ensembling is an interesting way to utilize for such a task.
4. The authors conduct many experiments to show the utility of the approach.
3. The added ablation studies in the paper are nice.
5. The paper is well-written and structured.

**Weaknesses:**

1. A clear connection to _selective prediction_ is missing. While the authors talk about _selective evaluation_, the concept is the same so it is better to make that link clear.
2. Related Work can benefit from more work on selective prediction and human subjectivity in LLMs, e.g.

Barbara Plank. 2022. The “Problem” of Human Label Variation: On Ground Truth in Data, Modeling and Evaluation. In Proceedings of the 2022 Conference on Empirical Methods in Natural Language Processing, pages 10671–10682, Abu Dhabi, United Arab Emirates. Association for Computational Linguistics.

Khurana, U., Nalisnick, E., Fokkens, A., & Swayamdipta, S. Crowd-Calibrator: Can Annotator Disagreement Inform Calibration in Subjective Tasks?. In First Conference on Language Modeling.

Bavaresco, A., Bernardi, R., Bertolazzi, L., Elliott, D., Fernández, R., Gatt, A., ... & Testoni, A. (2024). Llms instead of human judges? a large scale empirical study across 20 nlp evaluation tasks. arXiv preprint arXiv:2406.18403.

3. Cheaper models used for Cascading are still very costly (e.g. Mistral 7B)
4. The calibration set for human preferences is based on argmax of human judgments. Using argmax gets rid of disagreement signals, which is actually important information.

**Questions:**

How does IAA change for abstained samples with a lower target accuracy?

---

> ### Author Response · Authors · 2024-11-18
>
> We thank the reviewer for both positive and helpful comments! We are excited to see that you found our paper to tackle an important issue, well-written, and our experiments and ablations to be insightful! Below are responses to each of your suggestions:
>
> &nbsp;
>
> ### Stronger connection to Selective Prediction
> Thank you for the point.The objective of cascaded selective evaluation is indeed comparable to that of selection with guaranteed risk (SGR) [1] in the selective prediction literature, while the specific method differs in how to operationalize this guarantee. We formulate the selection of abstention thresholds as a multiple-testing problem, compared to Bonferroni Correction in SGR, allowing better coverage while providing the same guarantee **(Section 2.1)**. We also generalize the selective prediction guarantee to cascades of models, rather than a single judge model **(Section 2.3)**. We will clarify the link between selective evaluation and selective prediction, along with their guarantees.
>
> [1] Yonatan Geifman and Ran El-Yaniv. Selective classification for deep neural networks, 2017.
>
> &nbsp;
>
> ### Cheaper Judges are still costly?
> While the judge models with 7B parameters may not be the smallest possible model, we believe this is still a notable result, given the scale of judge models widely used in the literature. For example, in a recent paper that employs a suite of “small” juries instead of a single judge model [1], the smallest LLM employed as a jury is 35B (aside to other models used in parallel). Another paper [2] employs a 7B judge model as in our experiments, but the model has been fine-tuned with specialized synthetic dataset to improve its evaluation capability, while we employ an off-the-shelf model to maximize the framework’s utility and generalization.
>
> Even the strongest judge model GPT-4 fails to consistently align with humans, e.g. marking only 63.2% agreement in Auto-J **(Figure 5). On the contrary, our work aims to improve the alignment of LLM judges even without fully relying on the frontier models, thus making automatic evaluation more cost-effective and scalable.** That said, I’m excited to see even smaller judges than 7B-scale that stay competitive with their larger counterparts, which could make larger presence in the cascaded framework.
>
> [1] Pat Verga, Sebastian Hofstatter, Sophia Althammer, Yixuan Su, Aleksandra Piktus, Arkady Arkhangorodsky, Minjie Xu, Naomi White, and Patrick Lewis. Replacing judges with juries: Evaluating llm generations with a panel of diverse models, 2024.
>
> [2] Seungone Kim et al. Prometheus 2: An Open Source Language Model Specialized in Evaluating Other Language Models, In Proceedings of EMNLP 2024.
>
> &nbsp;
>
> ### Removal of disagreement signals
> While the evaluation of our framework is based on how well it predicts the majority preference label, this only serves for evaluation purposes (mainly due to the meta evaluation benchmarks evaluating majority alignment - e.g. both ChatArena and Auto-J provide majority human preference label only). **In fact, since our framework is objective-agnostic, it can simply be extended for individual alignment by changing what it’s supposed to guarantee (from agreement ratio with majority label to agreement ratio with individual preferences).** Specifically, one way to operationalize this is to collect a calibration set from a consistent set of human evaluators, and align the model judgement to match a specific annotator’s preference. In this regard, we believe Cascaded Selective Evaluation can serve as a useful tool for modeling personalized preferences with otherwise monolithic LLM judges, rather than removing useful individual preference information.
>
> &nbsp;
>
> ### What happens to Human IAA when we lower target agreement level?
> When we set target human agreement $1-\alpha$ to 0.8 instead of 0.9, the average human IAA for abstained samples goes from 0.815 to 0.806, i.e. only the harder-to-reconcile samples tend to be abstained by the models. This is due to the fact that when we lower the target agreement level, the evaluation coverage improves, hence the model tends to abstain less (only abstain when they sufficiently lack confidence).
>
> &nbsp;
>
> ### Related Works
> We appreciate the suggestions, these works are indeed relevant to improving reliability of LLM-based evaluation. We will include these works in our revision.
>
> &nbsp;
>
> We hope the responses above address your questions - thank you again for the constructive and positive feedback!

---

> > ### Comment · Reviewer_ZdoK · 2024-11-25
> >
> > Thank you for your response, I have increased my score!

---

> > > ### Author Response · Authors · 2024-11-25
> > >
> > > Thank you again for the positive feedback and adjusting your evaluation of our work!

---

### Official Review · Reviewer_8fUt · 2024-11-04

**Soundness:** 3
**Presentation:** 4
**Contribution:** 4
**Rating:** 10
**Confidence:** 5

**Summary:**

This paper proposes a selective evaluation framework to ensure that LLM-based evaluations align with human judgments to a defined degree of confidence. Selective Confidence Evaluation ensures that, instead of always trusting an LLM's judgment, the model evaluates its confidence in a given answer and abstains when unsure. Simulated Annotators leverage in-context learning to simulate diverse annotator perspectives, improving model calibration and reducing evaluation errors. Cascaded Selective Evaluation introduces multiple LLM judges used in a tiered manner, starting with a more cost-effective model and escalating to stronger models only if needed. The method achieves strong human alignment and reduces reliance on more costly models like GPT-4 by using smaller models when possible.

**Strengths:**

This paper provides clear methodological contributions to improving human-LLM alignments. I especially appreciate the cascaded selective evaluation approach, which explicitly considers the costs of using stronger/more costly LLMs. Overall, this is an excellent paper with clear contributions and is well-executed.

**Weaknesses:**

This paper has already been very carefully executed, and I have only minor suggestions.

1.	Simulated annotators, despite their encouraging performance, will require significant processing time and resources as N and K increase. Given a substantial decrease in ECE, I believe that simulated annotators have enough value but it would be helpful for the readers if the authors can provide time and cost requirements compared to the canonical approaches. Furthermore, the authors may want to invest a bit more space in explaining the algorithm of simulated annotators including exactly how they handled in-context learning.

2.	The outperformance of cascaded selective evaluation is somewhat expected because the gist of the framework is to exclude the observations where annotator disagreement is expected. In my view, the model does a good job of identifying these hard-to-reconcile observations. A fairer comparison would be to exclude the abstained observations from the baseline models and see how their performance is affected. For example, if you maintain the same coverage for all other methods in Table 2 and Table 3, what would their empirical human agreement look like? Of course, empirically this cannot be achieved without your model – but as you show in Section 3.3, some systematic characteristics can predict abstention. One can simply devise an alternative strategy that can identify these hard-to-reconcile samples and run a multi-agentic approach on them.

3.	I think the most striking result is in Table 7. It is surprising that the weaker cascades achieve an empirical human agreement of 80.3%. Although their coverage is 68.3%, this is indeed a remarkable result. Given this, I would try to do more with the abstained samples. Almost 30% of the sample is now abstained and running the most advanced model on those will not only increase your ultimate coverage and success rate. The relative API cost will be substantially lower than using only GPT4.

4.	Relatedly, the relative API cost of the model is largely dependent on the difficulty of your task. As seen in Tables 2, 3, and 8, the evaluator composition is heterogeneous across different tasks. When a smaller LLM can handle the majority of tasks, the overall task is likely to be cheaper. When a larger LLM needs to intervene, the overall task will be more expensive. I would caveat this (and perhaps show the range of relative API cost using some simulated evaluator compositions) in Section 3.6.

5.	It is a minor thing, but is there any reason that you chose 1-a =0.9 in Table 2, 1-a=0.85 in Table 3, and 1-a=0.8 in Table 7? I would personally prefer the three thresholds to be consistent throughout the paper.

Overall, it was an interesting read. Great paper.

**Questions:**

See above.

---

> ### Author Response · Authors · 2024-11-18
>
> We sincerely thank the reviewer for the positive feedback and great suggestions ! We are glad that you found our work to have clear methodological contributions and our experiments to be very carefully executed. Below, we address each of your comments:
>
> &nbsp;
>
> ### Cost comparison between confidence estimation methods
> Thank you for the comment. Following your suggestion, we compared the resource requirements of Simulated Annotators and canonical methods on AlpacaEval. We focus on the cost requirement (as measured by relative API cost), since the majority of considered methods can be parallelized to generate multiple generations with LLM – e.g. Semantic Set, Semantic Entropy – and thus their wall clock runtime wouldn’t be significantly different from Predictive Probability.
>
> |                              |   ECE ↓   |  AUROC ↑  | Relative API Cost |
> |------------------------------|:---------:|:---------:|:-----------------:|
> | Predictive Probability       |   0.374   |   0.457   |       1.000       |
> | Predictive Probability (CoT) |   0.292   |   0.527   |       20.376      |
> | Verbalized Confidence        |   0.414   |   0.490   |       2.215       |
> | Semantic Set                 |     -     |   0.513   |       20.376      |
> | Semantic Entropy             |     -     |   0.572   |       20.376      |
> | Simulated Annotators (N=K=3) |   0.089   |   0.613   |       3.513       |
> | Simulated Annotators (N=K=5) | **0.075** | **0.632** |       6.910       |
>
> The results are as shown above. Simulated Annotators under both settings outperform all canonical methods in terms of ECE and AUROC, while incurring significantly less cost than more sophisticated methods (Predictive Probability (COT), Semantic Set, Semantic Entropy). We’d like to also emphasize that while calibrating Simulated Annotators does make a tradeoff between calibration and cost compared to Predictive Probability (or zero-shot evaluating with LLM Judge with considering their confidence), the higher cost can often be offset under the cascaded framework by dynamically selecting cheaper judge to evaluate the sample.
>
> &nbsp;
>
> ###  Controlling coverage for baselines
> To remove the effect of confounders, for all baselines in Table 2 and 3, we used the same cascades of models and the same confidence measure (Simulated Annotators) as Cascaded Selective Evaluation — the only difference between our method and the considered baselines is in how they decide the abstention policy $\hat{\lambda}$ (and therefore the coverage of each LLM judge). Therefore, if we fix the coverage in a baseline with the value we obtained via Cascaded Selective Evaluation (e.g. in Table 2, enforcing Mistral-7B to evaluate 28.3% * 55.7% of the samples, GPT-3.5 to evaluate 28.2% * 55.7% of the samples, …), the baseline would be essentially equivalent to Cascaded Selective Evaluation.
>
> **The results on Table 2 and 3 show that even with the same judge models and confidence measure, the reliability of evaluation can be drastically different depending on how we set the abstention policy.** Unlike the baselines, Cascaded Selective Evaluation precisely controls $\lambda$ that guarantees target human agreement in the unseen set of samples.
>
> That said, we are excited to see future improvements on Cascaded Selective Evaluation that adopt even better confidence measures and better cascades of judge models. As you noted, given that the machine abstentions correlate with the subjectivity perceived by humans, perhaps a future work could explore modeling subjectivity with a multi-agentic system to better elicit hard-to-reconcile samples.
>
> &nbsp;
>
> ### Going further with abstained samples
> Thank you for the great suggestion! Below, we followed your suggestion to further augment the weaker cascades by adding GPT-4-turbo as the last judge.
>
> |                           | Empirical Human Agreement | Coverage | Guarantee Success Rate | Relative API Cost |
> |:-------------------------:|:-------------------------:|:--------:|:----------------------:|:-----------------:|
> |     *weaker cascades*     |            80.3           |   68.3   |          90.8          |       0.126       |
> | *weaker cascades + GPT-4* |            80.4           |   78.2   |          90.6          |       0.192       |
>
> Notably, the augmented cascades lead to substantially better coverage while providing the same guarantee as before. At the same time, the cost is still much less than relying solely on GPT-4 (1.000).

---

> ### Author Response · Authors · 2024-11-18
>
> ### Does the composition of judges correlate with the difficulty of the task?
> Yes, intuitively if the evaluation task is hard (e.g. evaluating generations for hard reasoning prompts), then the task would generally require stronger judges to better align with the ground truth. While this is generally true, the real challenge lies in that it is hard to predict (in advance) how strong of a judge we need to use to reliably evaluate a given task. In this sense, the core contribution of our framework is that it provides a way to select when to trust which judge model, while not worrying about the precision of the evaluation.
>
> In an ideal scenario, the evaluation wouldn’t incur any cost for API calls, as Mistral-7B would evaluate all samples without abstention. The worst case scenario for the cost of model cascades would be equivalent to using GPT-4 for full evaluation, so long as the baseline uses the same prompting strategy. However these scenarios are extreme, and most results will interpolate between the two, as shown in the results in **Table 2, 3, and 8**. We will include the discussion on the relationship between task difficulty and judge composition in Section 3.6.
>
> &nbsp;
>
> ### Consistent $1-\alpha$
> Thank you for notifying this, the target accuracy in the tables were chosen based on the difficulty of the evaluation task, as measured by GPT-4’s average human agreement level in each dataset. We think this way best compares the results against relying on GPT-4 in each benchmark, the most widely adopted evaluation scheme in the literature. However, we also detailed results across different levels of $1-\alpha$ in each accompanied figure **(right side of Figure 3, 4 and 5)**.
>
> |                   | Empirical Human Agreement | Coverage | Guarantee Success Rate | Relative API Cost |
> |:-----------------:|:-------------------------:|:--------:|:----------------------:|:-----------------:|
> | *weaker cascades* |            85.7           |   59.5   |          90.5          |       0.163       |
>
> Here, we additionally report the results for weaker cascades in Table 7 but with $1-\alpha = 0.85$, consistently with the setup of main experiments on ChatArena in Table 3.
>
> &nbsp;
>
> Again, thank you for your positive and constructive suggestion! We will clarify the rationale behind our choice of \alpha in a footnote.

---

> > ### Comment · Reviewer_8fUt · 2024-11-18
> > **Response**
> >
> > Thank you for your detailed response.
> >
> > 1. Related to the cost comparison results, I would report them in an Appendix.
> > 2. Regarding augmenting the model with GPT4turbo, I like the results very much. The only concern that I had about the paper was its relatively low coverage. Future work will have to improve the coverage in general, but it seems that you have already taken an important step here. I would definitely include these results in an appendix as well.
> > 3. Also, please include a brief description of the relation between task difficulty and judge composition. This also relates to the abstention rates.
> >
> > This paper is now a complete piece with strong empirical evidence. I adjusted my recommendation accordingly.

---

> > > ### Author Response · Authors · 2024-11-18
> > >
> > > We will reflect these in the revised version of the paper. Again, we really appreciate your positive feedback and thoughtful comments!

---

### Meta-Review · Area_Chair_wcW3 · 2024-12-20

**Metareview:**

This paper proposes a framework called Cascaded Selective Evaluation for improving LLM-based evaluation with guaranteed human agreement. The authors demonstrate that their approach can achieve over 80% human agreement while being more cost-effective than relying solely on frontier models like GPT-4.

Strengths: It addresses a critical need for reliable automated evaluation. Theoretical guarantees and empirical results both strongly support the proposed method.

Weaknesses: only validated on pairwise preference evaluation tasks, more qualitative analysis of abstention patterns might also be helpful

Overall this paper is a clear acceptance with all reviewers agreeing on the contribution of the paper. It makes significant methodological and practical contributions to improving LLM-based evaluation. The theoretical framework is sound, and the empirical results demonstrate clear improvements over existing approaches.

**Additional Comments On Reviewer Discussion:**

The paper received thorough discussion during the rebuttal period, the authors provided detailed and substantive responses to all major concerns. Three reviewers increased their scores after the rebuttal accordingly.

---

### Decision · Program_Chairs · 2025-01-22

Accept (Oral)